# Diabetic Retinopathy and Hearing Loss: Results from the Fifth Korean National Health and Nutrition Survey

**DOI:** 10.3390/jcm10112398

**Published:** 2021-05-28

**Authors:** Yong Un Shin, Seung Hun Park, Jae Ho Chung, Seung Hwan Lee, Heeyoon Cho

**Affiliations:** 1Department of Ophthalmology, Hanyang University College of Medicine, Seoul 04763, Korea; yushin@hanyang.ac.kr (Y.U.S.); park1211sh@gmail.com (S.H.P.); 2Department of Otolaryngology-Head and Neck Surgery, Hanyang University College of Medicine, Seoul 04763, Korea; jaeho.chung.md@gmail.com

**Keywords:** diabetic retinopathy, hearing loss, Korean National Health and Nutrition Survey

## Abstract

We investigated the association between the severity of diabetic retinopathy (DR) and hearing loss based on vascular etiology. We used data from the Korean National Health and Nutrition Survey 2010–2012. Adults aged >40 years with diabetes were enrolled. Demographic, socioeconomic, general medical, noise exposure and biochemical data were used. Participants were classified into three groups: diabetes without DR, non-proliferative DR (NPDR), and proliferative DR (PDR); participants were also divided into two groups (middle age (40 ≤ age < 65 years) vs. old age (age ≥ 65 years)). The association between hearing loss and DR was determined using logistic regression analysis. A total of 1045 participants (*n* = 411, middle-aged group; *n* = 634, old-age group) were enrolled. Overall, the prevalence of hearing loss was 58.1%, 61.4%, and 85.0% in the no DR, NPDR, and PDR groups, respectively. After adjusting for confounding factors, the logistic regression model showed that there was no significant association between the prevalence of DR and hearing loss in the overall sample. However, the presence of PDR (OR 7.74, 95% CI 2.08–28.82) was significantly associated with hearing loss in the middle-aged group. Middle-aged people with diabetes may have an association between DR severity and hearing loss. The potential role of microvascular diseases in the development of hearing loss, especially in middle-aged patients, could be considered.

## 1. Introduction

Sensorineural hearing loss is one of the most common sensory abnormalities in elderly people, and its pathophysiology remains unknown. Old age, male sex, history of noise exposure, and vascular risk factors (diabetes mellitus (DM), hypertension, and dyslipidemia) are known risk factors [1,2]. Among the vascular factors, it is believed that hearing loss may occur due to the cellular damage caused by reduced cochlear blood flow as a result of the stria vascularis alteration in the inner ear [1,3]. However, it is impossible to observe cochlear vasculature directly in vivo; therefore, it is difficult to understand the association between hearing loss and vascular causes.

The retina is a tissue in the eye through which the vascular status can be easily observed directly from outside the body in a non-invasive manner. Due to this characteristic, many studies have investigated the relationship between systemic disease states and retinal vascular conditions [4,5,6,7]. The retinal vasculature is used as a marker of microvascular disease in various systemic diseases such as hypertension, dyslipidemia, and diabetes. In particular, when diabetes is poorly controlled or has a long disease course, it progresses from the absence of diabetic retinopathy (DR) to either non-proliferative diabetic retinopathy (NPDR) or proliferative diabetic retinopathy (PDR) [8,9,10]. The association between the severity of DR and systemic problems is well known [11].

The retina and inner ear are known to be regulated by some of the same genes [12], and systemic diseases such as hypertension and diabetes are common risk factors for hearing loss [13,14] and microvasculopathy of the retina. Therefore, using DR as a reflection of the degree of vasculopathy, we hypothesized that the more severe the DR, the more severe the microangiopathy of the inner ear and the higher the frequency of hearing loss. In this study, we investigated the association between hearing loss and DR severity in patients with diabetes in Asian population.

## 2. Materials and Methods

### 2.1. Study Design and Population

The data for this study were obtained from the fifth Korea National Health and Nutrition Examination Survey (KNHANES V), conducted from 2010 to 2012 [15]. The KNHANES is a nationally representative cross-sectional survey conducted annually by the Korean Center for Disease Control and Prevention (KCDC). It began in 1998, and surveys were conducted in 1998, 2001, 2005, 2007–2009, and 2010–2012. Survey volunteers were selected using a stratified, multistage, probability sampling design. Details of the KNHANES sample recruitment strategy have been described elsewhere [16,17].

The KNHANES consists of three parts: a health interview survey, a health examination survey including ophthalmologic and otologic examinations, and a nutritional survey. The health interview survey was administered to all study participants, and ophthalmic and otologic examinations were performed simultaneously in the same mobile examination unit. The KNHANES was approved by the KCDC Institutional Review Board, and the survey adhered to the tenets of the Declaration of Helsinki; all participants provided written informed consent.

We included participants with DM aged 40 years and older in the analysis. DM was defined as a previous diagnosis of diabetes made by a physician, current treatment with insulin or oral hypoglycemic agents, and/or a fasting blood glucose level of ≥126 mg/dL. Participants were excluded from our analyses if they did not have gradable fundus photographs due to poor image quality or missing examination or interview data.

### 2.2. Data Collection

#### 2.2.1. Ophthalmic Examinations and Grading of DR

A digital non-mydriatic fundus camera (TRC-NW6S; Topcon, Tokyo, Japan) and a Nikon D-80 digital camera (Nikon, Tokyo, Japan) were used to obtain the fundus images. Seven standard field photographs were obtained for each eye of the participant after pharmacological pupil dilatation, as per the Early Treatment for Diabetic Retinopathy Study protocol, from 2010 to 2011 [18]. In contrast, in the last year of the KNHANES V (2012), fundus photography was performed for every participant with DM without pharmacological pupil dilatation [19].

DR was identified based on the presence of any characteristic lesion as per the Early Treatment for Diabetic Retinopathy Study severity scale: microaneurysms, hemorrhages, hard exudates, cotton wool spots, intraretinal microvascular abnormalities, venous beading, and new retinal vessels [9]. The prevalence of DR among individuals with DM was also estimated. Each fundus image was graded twice. First, preliminary grading was conducted onsite by ophthalmologists or ophthalmologic residents trained by the Korean Ophthalmologic Society. Retinal specialists with expertise in DR grading then performed detailed final grading using protocols from the Early Treatment for Diabetic Retinopathy Study.

#### 2.2.2. Assessment of Hearing Loss

The pure-tone air-conduction threshold was measured in a sound-treated booth using an automatic audiometer (GSI SA-203; Entomed AB, Malmo, Sweden). Trained otolaryngologists collected data independently for each ear at six frequencies: 0.5, 1.0, 2.0, 3.0, 4.0, and 6.0 kHz. All audiometric tests were performed under the supervision of an otolaryngologist. Hearing loss was defined as an average pure-tone air-conduction hearing thresholds in the better ear worse than 25 decibel hearing level (dBHL) at 0.5, 1, 2, and 4 kHz, according to the World Health Organization (WHO) definition. We classified hearing impairment into three categories based on severity: mild, moderate, and severe. Mild hearing impairment was defined as an unaided pure-tone hearing threshold level for the better ear of 25–39 dBHL and average hearing threshold levels at 0.5, 1, 2, and 4 kHz. Moderate hearing impairment was defined as an unaided pure-tone hearing threshold level for the better ear of 40–69 dBHL and average hearing threshold levels at frequencies of 0.5, 1, 2, and 4 kHz. Severe hearing impairment was defined as an unaided pure-tone hearing threshold level for the better ear of 70 dBHL or greater and average hearing threshold levels at frequencies of 0.5, 1, 2, and 4 kHz.

In this study, we have included several related factors that may affect hearing loss in patients with DM, as suggested in previous studies [20,21,22,23,24]. From the KNHANES databases, we collected data regarding various factors obtained through direct interviews using standardized questionnaires. For example, in the noise exposure questionnaire, there were questions on the “experience of using earphones in a noisy place”, “experience of exposure to noise in the workplace”, and “experience of instantaneous noise exposure”. If a participant answered “yes” to any of these items, they were considered to have been exposed to noise. Other variables including socioeconomic and medical status were defined by referring to the previous study using KNHANES databases [25].

### 2.3. Subgroup Analysis

Participants with diabetes were divided into three groups: diabetes without DR, diabetes with NPDR, and diabetes with PDR; all participants were divided into two groups based on an age cutoff of 65 years: middle-aged group (over 40 years old and under 65 years old) and old-age group (over 65 years old).

### 2.4. Statistical Analysis

The KNHANES sampling data were weighted by statisticians and statistical analysis was performed according to the statistical guidelines of the Korean Center for Disease Control and Prevention, an institution supervising the KNHANES. Demographic data and differences between the groups were analyzed using the Pearson’s chi-square test for categorical variables and an independent-sample t-test for continuous variables. The association between hearing loss and DR was determined using logistic regression analysis. Univariate logistic regression analyses were performed for analyzing the association between factors and hearing loss in all participants according to age group. Multivariate logistic regression analyses were performed for analyzing the association between hearing loss and DR after adjusting for all potential confounding factors. Complex sample analyses considering weights, stratified variables, and cluster variables were performed using SPSS software (version 22.0; IBM Corp., Armonk, NY, USA). All data are presented as the mean ± standard error, and statistical significance was set at a *p*-value of 0.05.

## 3. Results

### 3.1. Baseline Characteristics of Enrolled Participants

Of the 23,353 participants who underwent ophthalmic and otologic examinations, we excluded 21,572 participants because they were diabetes-free and aged <40 years. Of the remaining 1781 participants, 624 were excluded from the analysis because they did not have gradable fundus photographs due to poor image quality or were missing interviews, or laboratory data, and 112 participants with conductive hearing loss based on the otoendoscopic examination were excluded (Figure 1).

A total of 1045 participants were eligible for this study (411 participants in the middle-aged group and 634 participants in the old-aged group). Table 1 shows the detailed data of the enrolled participants according to age group. Overall, the mean age of all participants was 59.2 ± 0.4 years, and the prevalence of any hearing loss and DR were 57.9 ± 1.9% and 9.1 ± 1.1%, respectively. Table 2 shows demographic data stratified by age. The mean age of participants with hearing loss was significantly higher (63.4 ± 0.5 years) than that of participants without hearing loss (53.4 ± 0.5 years) (*p* < 0.001). Female sex was less common in participants with hearing loss than in those without hearing loss (40.2% ± 2.2% vs. 50.7% ± 3.1%, *p* = 0.007). Participants with hearing loss were more likely to have a lower education level (67.4% ± 2.4% vs. 54.8% ± 3.3%, high school or less, *p* < 0.001) and household income (63.2% ± 2.5% vs. 43.5% ± 3.1%, in the lower half, *p* < 0.001) than those without hearing loss. The prevalence of smoking was greater in participants with hearing loss than in those without hearing loss (56.2% ± 2.3% vs. 46.0% ± 3.3%, *p* = 0.013), while a higher body mass index (BMI) was observed in participants without hearing loss than in those with hearing loss (26.1 ± 0.3 vs. 25.0 ± 0.2 kg/m^2^, *p* = 0.001). Participants with hearing loss were more likely to have a history of noise exposure than those without hearing loss (39.0% ± 2.6% vs. 25.9% ± 2.7%, *p* = 0.001). Hypertension (63.8% ± 2.3% vs. 55.8% ± 3.2%, *p* = 0.028), chronic kidney disease (CKD) (8.9% ± 1.3% vs. 2.3% ± 0.8%, *p* < 0.001), and stroke history (4.9% ± 1.0% vs. 1.1 ± 0.5%, *p* = 0.001) were more frequently observed among participants with hearing loss than among those without hearing loss. Regarding laboratory data, the fasting blood sugar level (140.3 ± 2.2 vs. 149.4 ± 3.7 mg/dL), serum total cholesterol level (184.0 ± 2.1 vs. 192.0 ± 3.2 mg/dL), serum creatinine level (0.9 ± 0.0 vs. 0.8 ± 0.0 mg/dL), and estimated glomerular filtration rate (86.1 ± 1.1 vs. 92.2 ± 1.1, mL/min/1.73 m^2^) were significantly lower in participants with hearing loss than in those without hearing loss. There was no significant difference in the presence of DR (10.2% ± 1.5% vs. 7.5% ± 1.8%) between the two groups (*p* = 0.277). The proportions also did not differ according to the severity of DR between the two groups.

Comparing the demographic data of the exclusion and inclusion groups shows differences in age and household income; the group included in the study had a lower mean age (59.2 ± 0.4 years vs. 66.5 ± 0.3 years, *p* < 0.001) and higher household income (45.1% ± 2.0% vs. 33.8 ± 1.8%, in the upper half, *p* = 0.001) than the excluded group.

### 3.2. Association of Diabetic Retinopathy with Hearing Loss

Table 3 shows the results of univariate logistic regression analyses for the association between factors and hearing loss. In all participants, the analyses showed that older age (odds ratio (OR) 1.12, 95% confidence interval (CI) 1.10–1.14), male sex (OR 1.53, 95% CI 1.12–2.09), lower education level (OR 2.51, 95% CI 1.79–3.50), lower household income (OR 2.23, 95% CI 1.61–3.07), smoking (OR 1.50, 95% CI 1.09–2.08), noise exposure (OR 1.83, 95% CI 1.29–2.59), hypertension (OR 1.40, 95% CI 1.04–1.88), CKD (OR 4.19, 95% CI 1.86–9.45), previous stroke (OR 4.73, 95% CI 1.77–12.66), fasting blood glucose levels (OR 1.00, 95% CI 1.00–1.00), serum total cholesterol levels (OR 1.00, 95% CI 1.00–1.00), BMI (OR 0.92, 95% CI 0.88–0.96), serum creatinine levels (OR 3.19, 95% CI 1.14–8.95), and estimated glomerular filtration rate (OR 0.98, 95% CI 0.97–0.99) were significantly associated with hearing loss.

Factors such as age, sex, smoking status, and BMI were significantly associated with hearing loss in both the middle-aged and old-age groups. However, education levels, household income, noise exposure, and serum total cholesterol levels were significantly associated with hearing loss only in the middle-aged group, and serum creatinine levels were significantly associated with hearing loss only in the old-aged group. The presence of DR was not significantly associated with hearing loss in any age group. With respect to the severity of DR, there was no significant association between hearing loss and NPDR, while hearing loss was significantly more common in participants with PDR than in those without DR in the middle-aged group (OR 6.00, 95% CI 1.42–25.43) and in all participants (OR 4.24, 95% CI 1.54–11.79). The proportions of participants with hearing loss among participants with DR are shown in Figure 2.

According to multivariate logistic regression analysis, adjusting for all potential confounding factors, DR was not significantly associated with hearing loss in the entire group (adjusted OR 1.01, 95% CI 0.51–2.00). However, in the middle-aged group, hearing loss was more common in participants with PDR than in those without DR (adjusted OR 7.74, 95% CI 2.08–28.82) (Table 4).

## 4. Discussion

In this population-based, cross-sectional study, we assessed all risk factors for hearing loss in participants with DM that had shown significance in previously published population-based studies. There was no association between overall DR and sensorineural hearing loss for both age groups combined. However, after adjusting for potential associated factors, DR severity was significantly associated with hearing loss. In particular, the group with PDR had a 7.74 times higher risk of hearing loss than the group without DR in the middle-aged group.

The relationship between DR and hearing loss has long been studied. Jorgensen and Bush [26] reported that patients with PDR had twice the potential for developing hearing loss compared to those without DR. Friedman et al. [27] also reported an association between DR and hearing loss. Kurt et al. [21] divided retinopathy into three groups according to severity (no retinopathy, background retinopathy, and proliferative retinopathy) and analyzed the association with hearing loss. They showed a significant relationship between the severity of DR and the degree of hearing loss. They speculated that the same pathologic processes as those involved in microangiopathy that lead to DR can possibly lead to hearing loss. However, some studies have shown that DR and hearing loss are not related. Miller et al. [20] reported in a 1983 study that no correlation was found between hearing thresholds and DR of any severity. However, they did not evaluate the association between hearing loss and DR severity. In addition, this previous study was a single hospital-based study with a limited number of cases.

The Blue Mountains Eye Study was a population-based study of hearing loss [22], in which retinopathy and changes in retinal blood vessel structure were analyzed in association with hearing loss, and it was reported that only retinopathy was significantly associated with hearing loss in women. This study used Western data and did not target patients with diabetes. There was a population-based study including Asians, which reported that the prevalence of DR was not significantly different between the normal hearing group and hearing loss group [24]. They did not classify DR according to severity. In addition, in the group without DR, both participants with and without diabetes were included; therefore, the design was different from that of our study.

Ooley et al. [28] classified DR using the Early Treatment Diabetic Retinopathy Study (ETDRS) classification categories for DR and studied the relationship between the severity of DR and diabetic neurosensory hearing loss. After adjusting for the degree of diabetes control represented by HbA1c and creatinine levels, it was reported that the severity of DR and hearing loss were significantly associated. They subdivided DR into mild NPDR, moderate NPDR, and severe NPDR/PDR; unlike our study, they reported that there was a significant difference between no DR and mild-to-moderate NPDR in hearing loss severity. In that study, the mean age was 61.5 years in the mild NPDR group, 62.4 years in the moderate NPDR group, and 58.8 years in the severe NPDR/PDR group; these values were higher than that in the middle-aged group of this study (53.3 years). Considering that the risk of age-related hearing loss doubles 5 years after the age of 60 [29], and increases rapidly after the age of 60 years [30], it seems that age influenced the severity of hearing loss in the mild-to-moderate NPDR group, which led to results different from those of our study.

In a previous study [23] comparing groups with and without hearing loss among DM patients, fasting blood glucose levels (169.73 ± 74.82 mg/dL vs. 171.20 ± 40.69 mg/dL), BMI (28.95 ± 3.86 kg/m^2^ vs. 29.99 ± 4.22 kg/m^2^), and HbA1c levels (8.61 ± 2.02% vs. 8.38 ± 1.67%) did not show significant differences between the hearing loss and non-hearing loss groups. However, in our study, fasting blood glucose levels and BMI of participants with hearing loss were lower than those of participants without hearing loss.

Hearing loss in patients with DM is caused by chronic hyperglycemia causing vascular defects in the stria vascularis and vestibulocochlear nerves [1,3]. Hypertension, chronic kidney disease, and a stroke history further induce hearing loss due to the association between damaged blood vessels and hearing loss. Our study showed similar results. In particular, PDR was more strongly associated with hearing loss than NPDR, indicating that the more severe the retinal vascular changes, the greater is the relation with vascular damage to other organs. This is supported by previous studies, which reported that the risk of chronic kidney disease progression was higher in those with PDR than in those with NPDR and that PDR had an increased risk of incident cardiovascular disease [31,32].

Interestingly, in this study, the middle-aged group with PDR but not the old-age group with PDR was associated with hearing loss. Hearing loss is closely related to the aging process. We assume that diabetic vasculopathy has a greater influence on hearing loss than the aging process in the middle-aged group, while both aging and diabetic vascular changes simultaneously cause hearing loss in the older age group. The loss of a significant association between PDR and hearing loss in patients aged >65 years may be due to a potent aging factor that dilutes the effect of DR on hearing function.

This study had several limitations. First, it was impossible to define a causal relationship between risk factors and hearing loss because this was a cross-sectional study. A prospective longitudinal cohort study could help identify causal risk factors. Second, DR evaluation using fundus photos can underestimate the prevalence of DR because of the narrow viewing angle. Ultrawide fundus photography-based epidemiologic studies can provide more informative data in the future. Third, although risk factors were selected based on existing studies, there may have been factors that were not considered, leading to unexpected biases. Fourth, we did not divide the hearing loss group according to the frequency of hearing state. Fifth, the number of NPDR and PDR subjects was small. Ophthalmology and otolaryngology examinations were conducted simultaneously in the KNHANES for only three years since 2010. In addition, we excluded many subjects from the study if any of the health interview surveys, laboratory data, and fundus photography had problems. As a result, however, there were fewer subjects with DR in our study, showing a significant association with DR severity and hearing loss in middle-aged patients; however, additional studies are needed to prove a clearer association. Sixth, in this study, 35% (624 out of 1781) of the subjects were excluded because of incomplete data in interviews or laboratory findings and poor quality of fundus photography. The demographic data of the exclusion and inclusion groups show that the group included in the study had a lower mean age and higher household income than the excluded group. There may be socioeconomic factors influencing the selection of subjects, hence, care should be taken when interpreting the results. Nevertheless, this study had strengths, in that it used a large sample size. The KNHANES is a government-led survey conducted using a standardized process, and the data are representative of the South Korean population.

## 5. Conclusions

The retina is the only tissue that can be evaluated to directly observe the state of the internal blood vessels in the body. Retinopathy might reflect vascular pathologies in other organs. Therefore, studies have been conducted to evaluate systemic vascular conditions based on retinal vasculature status. To our knowledge, this is the first population-based study to investigate the association between DR severity and hearing loss in an Asian population. In our study, no association was found between the presence of DR and hearing loss in patients with diabetes at all ages. However, middle-aged people with diabetes may have an association between DR severity and hearing loss. The relationship between DR severity and hearing loss remains controversial; therefore, more evidence is needed to support this association. Further well-designed and longitudinal cohort studies on the relationship between DR severity and hearing loss will provide a more precise evidence.

## Figures and Tables

**Figure 1 jcm-10-02398-f001:**
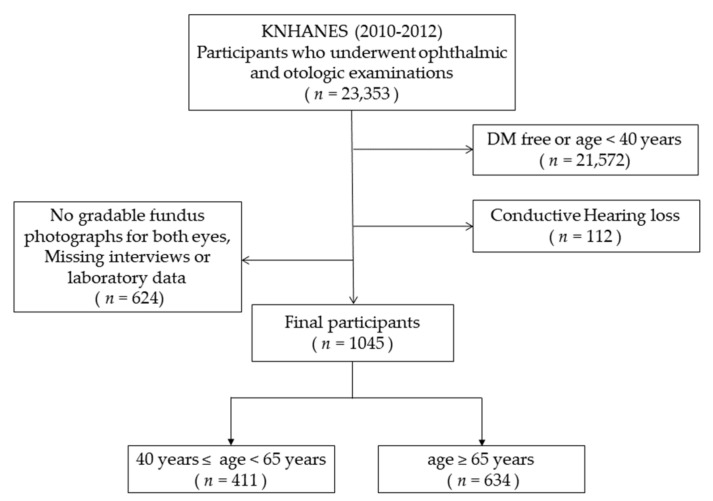
Flow chart of the selection of study participants. A total of 22,308 participants were excluded due to the following reasons: age <40 years, no diabetes (DM), no gradable fundus image for both eyes, missing interviews or laboratory data, and conductive hearing loss.

**Figure 2 jcm-10-02398-f002:**
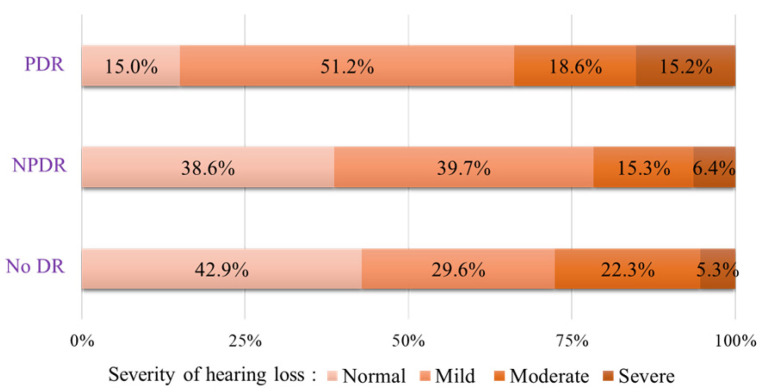
Distribution of the severity of hearing loss in patients with different degrees of diabetic retinopathy. Patients with PDR and NPDR were more likely to have hearing loss than patients without DR. PDR, proliferative diabetic retinopathy; NPDR, non-proliferative diabetic retinopathy; DR, diabetic retinopathy.

**Table 1 jcm-10-02398-t001:** Demographic and clinical characteristics of all participants and according to age group.

Associated Factors	Middle Age (40–64 years) (*n* = 411)	Old Age (≥65 years) (*n* = 634)	All Participants (*n* = 1045)
Age, years	53.3 ± 0.3	71.9 ± 0.3	59.2 ± 0.4
Sex, female, %	38.5 ± 2.3	57.7 ± 2.6	44.6 ± 1.8
Education level (greater than high school), %	52.7 ± 2.6	18.8 ± 2.1	42.0 ± 2.1
Household income (top 1/2), %	54.2 ± 2.7	25.7 ± 2.7	45.1 ± 2.0
Smoking, %	55.4 ± 2.6	44.1 ± 2.7	51.9 ± 1.9
Alcohol consumption, %	72.3 ± 2.1	64.5 ± 2.7	69.9 ± 1.7
Noise exposure, %	34.2 ± 2.5	31.9 ± 2.7	33.5 ± 2.0
Hypertension, %	53.6 ± 2.7	74.8 ± 2.4	60.4 ± 2.0
Chronic kidney disease, %	2.9 ± 0.8	13.2 ± 1.9	6.0 ± 0.8
Previous angina or MI, %	5.8 ± 1.2	14.2 ± 2.0	8.4 ± 1.0
Previous stroke, %	1.7 ± 0.6	6.8 ± 1.4	3.3 ± 0.6
Fasting blood glucose, mg/dL	149.6 ± 2.6	131.9 ± 2.1	144.3 ± 1.9
HbA1c, %	7.4 ± 0.1	7.2 ± 0.1	7.3 ± 0.1
Serum total cholesterol, mg/dl	188.9 ± 2.5	184.2 ± 2.4	187.5 ± 1.8
Body mass index, kg/m^2^	25.7 ± 0.2	24.9 ± 0.2	25.4 ± 0.2
Serum triglyceride, mg/dl	199.8 ± 13.0	163.0 ± 8.1	188.6 ± 9.1
BUN, mg/dL	15.4 ± 0.3	16.4 ± 0.3	15.7 ± 0.2
Creatinine, mg/dL	0.9 ± 0.1	0.9 ± 0.0	0.9 ± 0.0
eGFR, mL/min/1.73 m^2^	93.1 ± 1.0	78.7 ± 0.9	88.7 ± 0.8
Hearing loss, %	44.4 ± 2.5	86.7 ± 1.6	57.9 ± 1.9
Diabetic retinopathy, %	8.4 ± 1.4	10.5 ± 1.7	9.1 ± 1.1
No	91.6 ± 1.4	89.5 ± 1.7	90.9 ± 1.1
NPDR	7.3 ± 1.4	8.4 ± 1.6	7.7 ± 1.1
PDR	1.1 ± 0.4	2.1 ± 0.6	1.4 ± 0.3

All data are presented as mean ± standard error. MI, myocardiac infarction; BUN, blood urea nitrogen; DR, diabetic retinopathy; eGFR, estimated glomerular filtration rate; NPDR, non-proliferative diabetic retinopathy; PDR, proliferative diabetic retinopathy.

**Table 2 jcm-10-02398-t002:** Comparison of demographic data between participants with and without hearing loss.

Associated Factors	Hearing Loss (*n* = 678)				No Hearing Loss (*n* = 367)				*p*-Value
	Middle Age (40–64 Years)	Old Age (≥ 65 Years)	*p*-Value	All Participants	Middle Age (40–64 Years)	Old Age (≥65 Years)	*p*-Value	All Participants	
Age, years	55.2 ± 0.5	72.3 ± 0.3	<0.001	63.4 ± 0.5	51.7 ± 0.4	68.8 ± 0.5	<0.001	53.4 ± 0.5	<0.001
Sex, female %	26.3 ± 3.1	55.3 ± 2.9	<0.001	40.2 ± 2.2	48.1 ± 3.3	73.4 ± 6.2	0.002	50.7 ± 3.1	0.007
Education level, %									
High school or lower	54.3 ± 3.7	82.0 ± 2.3	<0.001	67.4 ± 2.4	41.6 ± 3.5	76.3 ± 5.9	<0.001	54.8 ± 3.3	<0.001
Greater than high school	45.7 ± 3.7	18.0 ± 2.3	<0.001	32.6 ± 2.4	58.4 ± 3.5	23.7 ± 5.9	<0.001	45.2 ± 3.3	<0.001
Household income, %									
Top 1/2	48.4 ± 3.8	24.2 ± 3.0	<0.001	36.8 ± 2.5	58.9 ± 3.3	35.3 ± 6.9	0.003	56.5 ± 3.1	<0.001
Lower 1/2	51.6 ± 3.8	75.8 ± 3.0	<0.001	63.2 ± 2.5	41.1 ± 3.3	64.7 ± 6.9	0.003	43.5 ± 3.1	<0.001
Smoking, %	64.3 ± 3.8	47.0 ± 2.9	0.001	56.2 ± 2.3	48.3 ± 3.6	25.8 ± 6.3	0.005	46.0 ± 3.3	0.013
Alcohol consumption, %	75.3 ± 3.0	65.7 ± 2.9	0.020	70.8 ± 2.1	69.9 ± 3.0	56.7 ± 7.0	0.084	68.6 ± 2.7	0.515
Noise exposure, %	45.3 ± 3.9	32.1 ± 3.1	0.007	39.0 ± 2.6	25.4 ± 2.9	30.8 ± 6.6	0.432	25.9 ± 2.7	0.001
Hypertension, %	53.3 ± 3.7	75.2 ± 2.7	<0.001	63.8 ± 2.3	53.9 ± 3.5	72.7 ± 6.2	0.014	55.8 ± 3.2	0.028
Chronic kidney disease, %	4.4 ± 1.5	14.1 ± 2.2	0.001	8.9 ± 1.3	1.6 ± 0.8	7.8 ± 4.1	0.02	2.3 ± 0.8	<0.001
Previous angina or MI, %	7.2 ± 2.0	13.7 ± 2.2	0.039	10.3 ± 1.5	4.6 ± 1.5	17.2 ± 5.5	0.003	5.9 ± 1.4	0.053
Previous stroke, %	2.8 ± 1.3	7.2 ± 1.6	0.051	4.9 ± 1.0	0.7 ± 0.4	4.1 ± 2.7	0.027	1.1 ± 0.5	0.001
Fasting blood glucose, mg/dL	147.0 ± 3.6	132.4 ± 2.3	0.001	140.3 ± 2.2	151.7 ± 4.0	129.1 ± 4.3	<0.001	149.4 ± 3.7	0.039
HbA1c, %	7.3 ± 0.1	7.2 ± 0.1	0.193	7.3 ± 0.1	7.5 ± 0.1	7.1 ± 0.1	0.048	7.5 ± 0.1	0.172
Serum total cholesterol, mg/dL	183.4 ± 3.1	184.7 ± 2.8	0.745	184.0 ± 2.1	193.2 ± 3.5	181.2 ± 5.1	0.054	192.0 ± 3.2	0.040
BMI, kg/m^2^	25.2 ± 0.3	24.7 ± 0.2	0.862	25.0 ± 0.2	26.1 ± 0.3	26.1 ± 0.5	0.973	26.1 ± 0.3	0.001
Serum triglycerides, mg/dL	190.5 ± 9.0	165.2 ± 8.8	0.057	178.9 ± 6.0	207.1 ± 22.0	149.6 ± 14.0	0.029	201.3 ± 19.8	0.283
BUN, mg/dL	15.6 ± 0.3	16.6 ± 0.3	0.025	16.1 ± 0.2	15.2 ± 0.4	15.6 ± 0.8	0.641	15.2 ± 0.4	0.050
Creatinine, mg/dL	0.9 ± 0.0	0.9 ± 0.0	0.189	0.9 ± 0.0	0.8 ± 0.0	0.8 ± 0.0	0.797	0.8 ± 0.0	0.003
eGFR, mL/min/1.73 m^2^	92.9 ± 1.6	78.2 ± 1.1	<0.001	86.1 ± 1.1	93.3 ± 1.1	82.0 ± 2.4	<0.001	92.2 ± 1.1	<0.001
Diabetic retinopathy, %	9.5 ± 2.1	10.9 ± 1.9	0.600	10.2 ± 1.5	7.5 ± 2.0	7.4 ± 4.0	0.973	7.5 ± 1.8	0.277
No	90.5 ± 2.1	89.1 ± 1.9	0.848	89.8 ± 1.5	92.5 ± 2.0	92.6 ± 4.0	0.285	92.5 ± 1.8	0.143
NPDR	7.5 ± 2.0	8.8 ± 1.8		8.1 ± 1.4	7.2 ± 1.9	5.4 ± 3.9		7.0 ± 1.8	
PDR	2.0 ± 0.8	2.1 ± 0.7		2.1 ± 0.5	0.3 ± 0.2	1.9 ± 1.2		0.5 ± 0.2	

All data are presented as mean ± standard error. BUN, blood urea nitrogen; DR, diabetic retinopathy; eGFR, estimated glomerular filtration rate; NPDR, non-proliferative diabetic retinopathy; PDR, proliferative diabetic retinopathy.

**Table 3 jcm-10-02398-t003:** Univariate logistic regression analyses for the association between factors and hearing loss in all participants and according to age group.

Associated Factors	Middle Age (40–64 Years)		Old Age (≥65 Years)		All Participants	
	OR (95% CI)	*p*-Value	OR (95% CI)	*p*-Value	OR (95% CI)	*p*-Value
Age (per 1-year increase)	1.09 (1.06–1.13)	<0.001	1.24 (1.13–1.36)	<0.001	1.12 (1.10–1.14)	<0.001
Sex, (male vs. female)	2.56 (1.70–4.00)	<0.001	2.22 (1.14–4.35)	0.019	1.53 (1.12–2.09)	0.008
Education (high school or less vs. greater than high school)	1.67 (1.11–2.50)	0.014	1.41 (0.71–2.86)	0.327	2.51 (1.79–3.50)	<0.001
Household income (lower 1/2 vs. top 1/2)	1.51 (1.03–2.27)	0.033	1.72 (0.85–3.45)	0.134	2.23 (1.61–3.07)	<0.001
Smoking, %	1.93 (1.24–3.01)	0.004	2.55 (1.31–4.94)	0.006	1.50 (1.09–2.08)	0.013
Alcohol consumption, %	1.32 (0.86–2.01)	0.206	1.46 (0.81–2.64)	0.206	1.11 (0.81–1.53)	0.515
Noise exposure, %	2.44 (1.58–3.79)	<0.001	1.06 (0.53–2.11)	0.866	1.83 (1.29–2.59)	0.001
Hypertension, %	0.98 (0.67–1.43)	0.914	1.14 (0.57–2.28)	0.712	1.40 (1.04–1.88)	0.028
Chronic kidney disease, %	2.73 (0.83–9.06)	0.100	1.94 (0.59–6.33)	0.272	4.19 (1.86–9.45)	0.001
Previous angina or MI, %	1.60 (0.66–3.90)	0.299	0.77 (0.32–1.86)	0.558	1.83 (0.99–3.40)	0.056
Previous stroke, %	4.02 (0.91–17.70)	0.066	1.78 (0.44–7.31)	0.421	4.73 (1.77–12.66)	0.002
Fasting blood glucose, mg/dL	1.00 (1.00–1.00)	0.389	1.00 (1.00–1.01)	0.516	1.00 (1.00–1.00)	0.030
HbA1c, %	0.94 (0.81–1.09)	0.400	1.04 (0.82–1.30)	0.768	0.91 (0.81–1.03)	0.148
Serum total cholesterol, mg/dL	1.00 (1.00–1.00)	0.030	1.00 (1.00–1.01)	0.561	1.00 (1.00–1.00)	0.030
Body mass index, kg/m^2^	0.94 (0.90–0.99)	0.020	0.88 (0.80–0.97)	0.009	0.92 (0.88–0.96)	<0.001
Serum triglyceride, mg/dL	1.00 (1.00–1.00)	0.370	1.00 (1.00–1.01)	0.408	1.00 (1.00–1.00)	0.125
BUN, mg/dL	1.02 (0.97–1.07)	0.451	1.05 (0.95–1.15)	0.332	1.04 (0.99–1.09)	0.116
Creatinine, mg/dL	2.30 (0.77–6.84)	0.134	6.08 (1.09–33.87)	0.039	3.19 (1.14–8.95)	0.028
eGFR, mL/min/1.73 m^2^	1.00 (1.00–1.01)	0.834	0.99 (0.97–1.01)	0.166	0.98 (0.97–0.99)	0.001
Diabetic retinopathy, %	1.29 (0.62–2.70)	0.496	1.55 (0.45–5.30)	0.487	1.40 (0.76–2.57)	0.279
No	Reference	0.052	Reference	0.796	Reference	0.019
NPDR	1.07 (0.48–2.38)		1.69 (0.35–8.03)		1.20 (0.62–2.31)	
PDR	6.00 (1.42–25.43)		1.15 (0.29–4.68)		4.24 (1.54–11.79)	

BUN, blood urea nitrogen; DR, diabetic retinopathy; eGFR, estimated glomerular filtration rate; MI, myocardial infarction; NPDR, non-proliferative diabetic retinopathy; PDR, proliferative diabetic retinopathy.

**Table 4 jcm-10-02398-t004:** Multivariate logistic regression analyses for the association between hearing loss and diabetic retinopathy after adjusting for all potential confounding factors.

Diabetic Retinopathy	Age Stratification ^†^				All Participants *	
	Middle Age (40–64 Years)		Old Age (≥65 Years)			
	OR (95% CI)	*p*-Value	OR (95% CI)	*p*-Value	OR (95% CI)	*p*-Value
Presence of DR	0.97 (0.43–2.20)	0.944	1.61 (0.40–6.48)	0.502	1.01 (0.51–2.00)	0.976
Severity of DR		0.008		0.709		0.195
No	Reference		Reference		Reference	
NPDR	0.77 (0.32–1.85)		1.96 (0.33–11.72)		0.88 (0.42–1.82)	
PDR	7.74 (2.08–28.82)		0.73 (0.17–3.09)		2.99 (0.87–10.29)	

* Overall analysis adjusted for confounding factors (age, sex, education, household income, smoking status, alcohol drinking status, noise exposure, hypertension, chronic kidney disease, previous angina or MI, previous stroke, fasting blood glucose, HbA1c, serum total cholesterol, body mass index, serum triglyceride, BUN, creatinine, and eGFR). ^†^ Age-stratified analysis adjusted for all confounding factors (age, sex, education, household income, smoking status, alcohol drinking status, noise exposure, hypertension, chronic kidney disease, previous angina or MI, previous stroke, fasting blood glucose, HbA1c, serum total cholesterol, body mass index, serum triglyceride, BUN, creatinine, and eGFR). CI, confidence interval; DR, diabetic retinopathy; eGFR, estimated glomerular filtration rate; OR, odds ratio; NPDR, non-proliferative diabetic retinopathy; PDR, proliferative diabetic retinopathy.

## Data Availability

The data presented in this study are available on request from the corresponding author.

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
