# Peer review of "Diabetic Retinopathy and Hearing Loss: Results from the Fifth Korean National Health and Nutrition Survey"

_jcm, 2021, doi:10.3390/jcm10112398_

Round 1
Reviewer 1 Report
The authors have responded to the issues raised in the original review and the quality of the manuscript has improved.
Author Response
We would like to convey our sincere thanks for taking the time to review our manuscript, and we greatly appreciate the opportunity that we have been given to further revise the manuscript. Thank you very much.
Reviewer 2 Report
SUMMARY
This population-based paper evaluates the relationship between hearing loss and diabetic retinopathy. Given the high prevalence of both DM and hearing loss, this is an important public health issue that also has implications for clinical practice. There were some places where additional detail would be ideal. The primary concern I have is regarding the lack of sample weight application as I discuss in more detail below. In addition, the use of an all-frequency PTA is somewhat surprising and in contrast to most published literature related to this topic. Why not use a standard (0.5, 1, 2, and maybe 4 kHz) PTA or a low and high frequency PTA? Overall, this study will add an important contribution to the literature, but the issues below should be addressed prior to publication.
ABSTRACT
This statement is a bit confusing: “There was no significant association between the prevalence of DR and hearing loss in patients with diabetes at all ages.” Perhaps it should read “There was no significant association between the prevalence of DR and hearing loss for middle and older aged subjects combined.” Or “In the overall sample, there was not a significant association between DR and hearing loss.”
Page 1, Lines 20-22: If the main point of this paper is to explore DR and HL, it might not be necessary to include results on these other factors in the Abstract. Instead (unless in contrast to the Journal’s style preference), the authors might consider providing OR (95% CI) for the most relevant findings.
INTRODUCTION
The Introduction was well written but did not capture the novelty of exploring this topic in the Asian population, which comes up for the first time in the Discussion/Conclusion. Perhaps a brief statement about this novelty would “sell” the paper more in the Introduction.
METHODS
Are the 2010-12 data from one cycle or two?
The Korean NHANES, like the US NHANES, requires incorporation of sample weights in statistical analysis to account for the complex sample design. Unless I missed it, this paper does not apply sample weights which may results in biased estimates. Analysis should be redone to account for this complexity. If it was already done, please state that. For a greater discussion of this issue see: Kim Y, Park S, Kim NS, Lee BK. Inappropriate survey design analysis of the Korean National Health and Nutrition Examination Survey may produce biased results. J Prev Med Public Health. 2013;46(2):96-104. doi:10.3961/jpmph.2013.46.2.96
What drove the decision to exclude a reference group of nondiabetics?
Page 2, lines 71-72: Should this statement be “self-report of physician diagnosis?” re: “DM was defined as a previous diagnosis of diabetes made by a physician,…”. Or were these diagnoses extracted from medical records?
Page 3, line 96: ‘soundproof’ is the appropriate term for an anechoic chamber. I think a more appropriate term here would be ‘sound-treated’
Page 3, lines 122-123: regarding this statement: “Multivariate logistic regression analyses were performed for analyzing the association between hearing loss and DR after adjusting for all potential confounding factors.” Please state the specific factors here.
The PTA is an average of all thresholds (0.5, 1, 2, 3, 4, 6 kHz). Most prior reports use a low and high frequency PTA or even just a “standard” PTA of 0.5, 1, 2, 4. What is the rationale for averaging all thresholds from all frequencies for one PTA?
Exclusions were not stated in text. For example, were participants required to pass an otoscopic or tympanometric examination? Were asymmetric hearing losses excluded? What about individuals with otologic pathology (e.g., otitis media, acoustic neuroma, etc.)? Figure 1 suggests persons with CHL were excluded. How was CHL defined? I recommend adding that information to the text to supplement Fig. 1.
A comparison of included and excluded participants was placed in the last paragraph of the Discussion. Consider stating this information earlier in the manuscript. Section 3.1 might be a good place for this information.
RESULTS
Tables include hypertension. How was hypertension defined?
Table 1: Why not report standard error? Regardless, clarify in the table legend or table column header that these data are mean (SD or SEM).
Table 1: I am not sure if p-values are needed here given that this table simply reflects the demographic composition of the study population.
Table 1: Does not include sample size; consider adding N.
Table 3: reports noise exposure. How was noise exposure defined?
Table 4: Why are p-values reported for the reference group but not NPDR or PDR? This is especially surprising for the middle-aged group because the OR (95% CI) suggests P<0.05.
Table 4: If factors such as income and age (for example) were not associated with hearing loss in univariate analysis (Table 3) then why adjust for them in Table 4?
CONCLUSIONS
Page 8, Lines 226-228: The first sentence of the Discussion is rather hard to follow; suggest reworking
In general, the terminology “all ages” is a bit misleading. Perhaps something like this would be clearer: “There was no association between overall DR and sensorineural hearing loss in diabetic patients for both age groups combined” or “There was no association between overall DR and sensorineural hearing loss in diabetic patients before age stratification.”
Page 9, Lines 264-265: This statement (“Considering that the risk of age-related hearing loss doubles after 5 years[27]…”)was hard to follow. The risk doubles after 5 years starting at what age? For example, the risk between 30 and 35 might differ from the risk between 60 and 65y. Please clarify.
Page 9, Lines 273-274: This finding is rather surprising. What might explain it? “However, in our study, fasting blood glucose levels and BMI of participants with hearing loss were lower than those of participants without hearing loss.”
Page 9, Lines 275-276: Is this statement referring to CN VIII? If not, which other nerves are involved? “Hearing loss in patients with DM is caused by chronic hyperglycemia causing vascular defects in the stria vascularis and nerves…”
Page 9, Line 298: What does this mean? Severity of hearing loss? “Fourth, we did not divide the hearing loss group according to the 298 frequency of hearing state”
----
CONFIDENTIAL COMMENTS TO EDITOR
This population-based study evaluates the relationship between hearing loss and diabetic retinopathy in an Asian population. Given the high prevalence of both DM and hearing loss, this is an important public health issue that also has implications for clinical practice. However, I do have significant concern about the lack of sample weights. If sample weights were used in this analysis, then this is a simple fix (clarify in text). However, if sample weights were not used, then the analysis needs to be redone to incorporate them so that the results are generalizable to the target population. If this is not done, it is my opinion that this is a fatal flaw that would merit manuscript rejection. This concern is what drove my rating of “low” for scientific soundeness. For further discussion on the topic see:
Kim Y, Park S, Kim NS, Lee BK. Inappropriate survey design analysis of the Korean National Health and Nutrition Examination Survey may produce biased results. J Prev Med Public Health. 2013;46(2):96-104. doi:10.3961/jpmph.2013.46.2.96
Aside from that, the rationale for defining hearing loss using a PTA that averages all test frequencies together is unclear, but this is not a fatal flaw if it can be rationalized in some way. Ideally, though, I think a low- and high-frequency analysis would be better given that the literature suggests frequency-specific associations between HL and some cardiometabolic risk factors.
Author Response
We would like to convey our sincere thanks for taking the time to review our manuscript, and we greatly appreciate the opportunity that we have been given to further revise the manuscript. We have made the changes recommended by the editor and reviewers.
Please see the attachment.

Round 2
Reviewer 2 Report
The revised version of this manuscript is much improved. My concerns/questions regarding the initial submission have been adequately addressed in this revision. I identified four minor remaining issues that should be addressed prior to publication.
(1) Section 2.2.2. Degree of hearing loss should use units ‘dB HL’ (e.g., on lines 107, 111, 115).
(2) Table 4, footnote: “All data are presented as mean +- SEM.” Aren’t these results odds ratios (95%CI) though? Unless I’m missing something, I think the “all data are presented..” sentence can be deleted.
(3) I recommend adding the definition of hypertension (systolic blood pressure of 140 mmHg or more, a diastolic bloodpressure of 90 mmHg or more, or taking drugs for hypertension) somewhere, even if it’s just in a table footnote.
(4) Similar to my hypertension comment – I recommend adding the definition of noise exposure (‘experience of usingearphones in a noisy place’, ‘experience of exposure to noise in the workplace’, and ‘experience of instantaneous noise exposure'. If a participant answered "yes" to any of these items, they were considered to have been exposed tonoise) somewhere in the methods or as a table footnote.
Author Response
Dear Editor & Reviewers:
We would like to convey our sincere thanks for taking the time to review our manuscript, and we greatly appreciate your detailed review. Thanks to your advice, we have been able to write a higher-quality manuscript. We have made the changes recommended by the editor and the reviewers.
(1) Section 2.2.2. Degree of hearing loss should use units ‘dB HL’ (e.g., on lines 107, 111, 115).
Those with only ‘dB’ written in the unit have been changed to ‘dBHL’ in the revised manuscript. (Page 3, lines 101, 104, 106 and 108)
(2) Table 4, footnote: “All data are presented as mean +- SEM.” Aren’t these results odds ratios (95%CI) though? Unless I’m missing something, I think the “all data are presented..” sentence can be deleted.
Thanks for pointing out the error. We have removed that content from the footnotes of Tables 3 and 4.
(3) I recommend adding the definition of hypertension (systolic blood pressure of 140 mmHg or more, a diastolic blood pressure of 90 mmHg or more, or taking drugs for hypertension) somewhere, even if it’s just in a table footnote.
We answered this next question.
(4) Similar to my hypertension comment – I recommend adding the definition of noise exposure (‘experience of using earphones in a noisy place’, ‘experience of exposure to noise in the workplace’, and ‘experience of instantaneous noise exposure'. If a participant answered "yes" to any of these items, they were considered to have been exposed to noise) somewhere in the methods or as a table footnote.
As requested, we have added the definitions of noise exposure. The definitions of other common covariables such as hypertension or diabetes have already been introduced in many previous studies; therefore, in this revised version, we have added the reference of our previous epidemiologic study where the definitions of variables were described in detail.
[Materials and Methods] (page 3, lines 111-119 in the revised manuscript)
In this study, we have included several related factors that may affect hearing loss in patients with DM, as suggested in previous studies. [20-24] From the KNHANES databases, we collected data regarding various factors obtained through direct interviews using standardized questionnaires. For example, in the noise exposure questionnaire, there were questions on the ‘experience of using earphones in a noisy place’, ‘experience of exposure to noise in the workplace, and ‘experience of instantaneous noise exposure'. If a participant answered "yes" to any of these items, they were considered to have been exposed to noise. Other variables, including socioeconomic and medical status, were defined by referring to the previous study using KNHANES data.[25]
[reference]
- Miller, J.J.; Beck, L.; Davis, A.; Jones, D.E.; Thomas, A.B. Hearing loss in patients with diabetic retinopathy. American Journal of Otolaryngology 1983, 4, 342-346, doi:https://doi.org/10.1016/S0196-0709(83)80021-0.
- Kurt, E.; Öztürk, F.; Günen, A.; Sadikoglu, Y.; Sari, R.A.; Yoldas, T.K.; Avsar, A.; Inan, Ü.Ü. Relationship of retinopathy and hearing loss in type 2 diabetes mellitus. Annals of Ophthalmology 2002, 34, 216-222, doi:10.1007/s12009-002-0026-4.
- Liew, G.; Wong, T.Y.; Mitchell, P.; Newall, P.; Smith, W.; Wang, J.J. Retinal microvascular abnormalities and age-related hearing loss: the Blue Mountains hearing study. Ear and hearing 2007, 28, 394-401, doi:10.1097/AUD.0b013e3180479388.
- Ashkezari, S.J.; Namiranian, N.; Rahmanian, M.; Atighechi, S.; Mohajeri-Tehrani, M.-R.; Gholami, S. Is hearing impairment in diabetic patients correlated to other complications? In J Diabetes Metab Disord, 2018; Vol. 17, pp 173-179.
- Kim, J.M.; Kim, S.Y.; Chin, H.S.; Kim, H.J.; Kim, N.R.; Epidemiologic Survey Committee Of The Korean Ophthalmological Society, O. Relationships between Hearing Loss and the Prevalences of Cataract, Glaucoma, Diabetic Retinopathy, and Age-Related Macular Degeneration in Korea. Journal of clinical medicine 2019, 8, doi:10.3390/jcm8071078.
- Shin, Y.U.; Lim, H.W.; Hong, E.H.; Kang, M.H.; Seong, M.; Nam, E.; Cho, H. The association between periodontal disease and age-related macular degeneration in the Korea National health and nutrition examination survey: A cross-sectional observational study. Medicine 2017, 96, e6418, doi:10.1097/md.0000000000006418.
Again, thank you so much for your help with our paper.
We also attached the authorship change form.
Best regards,
Heeyoon Cho, M.D. PhD, and Seung Hwan Lee, M.D. PhD, on behalf of the authors

This manuscript is a resubmission of an earlier submission. The following is a list of the peer review reports and author responses from that submission.
Round 1
Reviewer 1 Report
In this study, the association between diabetic retinopathy and hearing loss is investigated. Evidence of association between severity of diabetic retinopathy and hearing loss might indicate microangiopathy as potential patophysiological mechanism for development of hearing loss.
However, there are some issues that need to be addressed:
Materials and methods: During the first years of the survey, fundus images were obtained after pupil dilatation. However, during 2012 fundus photos were obtained without pupil dilatation. Did this have any impact on image quality?
Quite a large proportion of participants (624 out of 1781 = 35%) were excluded from the study due to non-gradable fundus photos due to poor image quality. Were these excluded individuals different in any respect from the included participants (age, sex, education, etc)? Authors should discuss potential impact of excluded population on results of the study.
Results: Table 1 presents demographic and clinical characteristics of all participants and according to age group. From this table it can be concluded that in total, there were 95 indiviuals (9% out of 1045) with diabetic retinopathy and only in total 15 individuals (1.4% out of 1045) with proliferative diabetic retinopathy. Authors need to discuss how the small number of individuals with diabetic retinopathy might affect the interpretation of their data.
Results: On page 4, lines 138-162 regarding comparison of demographic data between participants with and without hearing loss, a large proportion of data is also presented in table 2. Double presentation of data should be avoided.
Results/Discussion
In the discussion, lines 229-232, authors state that ”After adjusting for potential associated factors, diabetic retinopathy severity was significantly associated with hearing loss. In particular, the group with proliferative diabetic retinopathy had a 7.74-times higher risk of hearing loss than the group without diabetic retinopathy in the middle-aged group.”
I think this conclusion can be questioned, since the finding is based on 4 or 5 individuals with proliferative diabetic retinopathy in this specific age-group. In fact, on page 5 lines 160-162, authors state that there was no significant difference in the presence of diabetic retinopathy between the two groups (with and without hearing loss, respectively). Further, authors state (line 162) that the proportions did not differ according to the severity of diabetic retinopathy between the two groups. Using univariate logistic regression analysis the presence of diabetic retinopathy was not significantly associated with hearing loss in any group (lines 190-191). And finally, using multivariate logistic regression analysis, adjusting for all potential confounding factors, diabetic retinopathy was not significantly associated with hearing loss in the entire group (adjusted OR 1.01; lines 211-213). I think that a more appropriate conclusion would be that based on the whole group no association between diabetic retinopathy and hearing loss was found. Authors need to discuss in detail and better justify or alter their conclusion.
Language:
Introduction, lines 41-43, sentence starting: ”In particular, when diabetes is poorly…”. This sentence cannot be understood, please check for clarity.
Results, line 150: abbreviation CKD should be explained
Discussion, line 255: reference [26] should be Ooley et al.
Reviewer 2 Report
I add my review
